# Finger Millet Seed Coat—A Functional Nutrient-Rich Cereal By-Product

**DOI:** 10.3390/molecules27227837

**Published:** 2022-11-14

**Authors:** Oluwatoyin Oladayo Onipe, Shonisani Eugenia Ramashia

**Affiliations:** Department of Food Science & Technology, Faculty of Science, Engineering & Agriculture, University of Venda, Private Bag X5050, Thohoyandou 0950, South Africa

**Keywords:** finger millet bran, polyphenols, arabinoxylan, dietary fibre, anticholesterolemic, anti-cataractogenic

## Abstract

Finger millet (FM) is one of the little millets grown in Asia and Africa. Although still classified as an “orphan crop”, there is an increasing interest in the research of FM seed coat (FMSC), also known as bran. It houses 90% of the seed’s polyphenols and dietary fibre. The calcium and phosphorus content of FMSC is about 6- to 25-fold that of other cereals. FMSC is specifically beneficial for its polyphenols, arabinoxylans, phytates, and flavonoids content. Evidence of the hypoglycaemic, nephroprotective, hypocholesterolemic, and anti-cataractogenic effects of FMSC has been substantiated, thereby supporting the health claims and validating its nutraceutical potential for diabetics. This article discusses FMSC extraction and nutritional properties, focusing on arabinoxylan and polyphenols, their potential health benefits, and their application in food formulations. Although there is a dearth of information on using FMSC in food formulation, this review will be a data repository for further studies on FMSC.

## 1. Introduction

Millets are cereal grains of the family *Poaceae*. They are regarded as underutilized orphan crops and underutilized minor cereal crops and are majorly cultivated in Asia and Africa. Finger millet (*Eleusine coracana*) is one of the millet seeds cultivated on the African continent [1]. The seed coat comprises up to 20% of the total kernel weight, with rich content of phytates, minerals, phytochemicals, colour pigments and non-starchy polysaccharides [2]. The finger millet (FM) kernel is made up of three major layers, namely: seed coat, which accounts for 13–15%, the embryo (1.5–2.5%), and the endosperm (80–85%). FM is different from other millets due to its small size and the tight fusion of its seed coat to the aleurone layer, thereby leading to the cluster of nutrients within the seed coat/bran. The dietary fibre (11.5–19.6%) of FM is higher than other cereals such as oat, sorghum, foxtail millet and proso millet (Table 1). When compared to major grains such as wheat and rice, the consumption of FM elicits a substantially lower glycemic index—caused by the presence of polyphenols in it [2]. In calcium content, FM surpasses other cereals. FM has 6-, 8-, 12-, and 25-fold the calcium content of oats, pearl millet, wheat, and proso millet, respectively [3,4,5]. The United States National Academies consider finger millet a possible “super cereal,” as it is one of the most nutrient-dense of all the minor cereals [4]. There are several in-depth reviews of the nutritional properties, technological processing, and food use of finger millet [3,6,7,8]. However, there is only a small amount of research on the FM seed coat (FMSC). To our knowledge, there has been no review paper on FMSC. Therefore, this review compiles the research on the extraction, nutritional composition, health benefits and food use of the seed coat matter of FM.

## 2. Finger Millet Seed Coat (Bran)

The FM seed coat or bran is the by-product of millet grain milling and has a healthy amount of dietary fibre, minerals, and phytochemicals [6,9]. Normally, FM is milled along with its seed coat, where most nutrients, dietary fibre, and bioactive compounds are concentrated, thus offering nutritional and health benefits [10,11]. The FM wholemeal flour has a dark hue, thereby limiting its use in some bakery products owing to the low consumer appeal and the chewy, gritty texture [12]. To obtain a more acceptable refined millet flour with more potential for food formulation, the bran or seed coat is removed, constituting waste products that end up in landfills or are used as animal feed. The FMSC can be used to make composite flour for bakery products. The seed coat contains a significant number of tannins which accounts for the astringency associated with FM-based products. FM-rich foods alleviate diabetes, obesity, and related co-morbidities. An FMSC-rich diet has been shown to reduce inflammation, improve plasma lipid profile, reduce oxidative stress, control the expression levels of numerous obesity-related genes, and increase beneficial gut bacteria such as *Roseburia, Bifidobacteria* and *Lactobacillus* spp [2,13].

### 2.1. Extraction of Finger Millet Seed Coat

Two main methods of extraction are explained in the literature by Malleshi [14] and Shobana et al. [15]. The aim of the former was to produce decorticated FM grains (Figure 1A). The seed coat (SC) was a by-product of this process. Most cereal grains are easily de-branned, but due to the small size and tight adherence of FM bran to the endosperm, a hydrothermal pre-treatment was needed to extract the SC, leaving an intact endosperm-rich decorticated FM grain [14]. The aim of the method of Shobana et al., on the other hand, was to extract the seed coat directly [5,15]. The process of de-branning starts with tempering the grains with 5–10% *w/v* water for ten to thirty minutes. The grains are milled and sieved three times to obtain the tailings retained on the sieve. The tailings are washed, dried, and pulverised to obtain the bran (Figure 1B). With FM still cultivated on a minor scale in Africa, the SC is not commercially available like the bran of other cereals such as oat and wheat. An increase in the cultivation and food application of FM will potentially lead to the commercial availability of FMSC in stores.

### 2.2. Effect of Grain Pre-Treatments on Nutrient Composition of Finger Millet Seed

The purpose of pre-treatments is to weaken or degrade the cellulosic-phenolic network in plants to release and recover phenolic compounds while also disrupting the structure of the plant cell wall [16]. Bran of other cereals, including wheat [16,17], oat [18], corn [19], rice [20], foxtail millet [21], and pearl millet [22] have been pre-treated to improve their functionality and nutritional profile [23]. However, in the case of FM, the documented pre-treatment attempts have been on the grains, followed by bran extraction analysis. This is mostly due to the size of the seeds, leading to low bran yield. Hithamani and Srinivasan [24] assessed the effect of roasting, sprouting, microwave heating, open-pan boiling and pressure cooking on the phenolic compounds of FM wholemeal. They found out that roasting and sprouting increased the bioaccessibility of FM polyphenols. Krishnan et al. [12] utilised two pre-treatments: malting and hydrothermal treatments and characterised the physicochemical composition. They observed that malting increased protein, soluble and total dietary fibre and reduced the fat, carbohydrate, ash, calcium, phosphorus and phytate content. On the other hand, hydrothermal treatment reduced protein, ash and phosphorus while increasing fat, calcium, and insoluble and total dietary fibre (Table 2). These nutrient variations are due to pre-treatment differences, especially the germination process in the malted FMSC, which reduced the carbohydrate content [12]. During germination, amylase in the endosperm is activated, catalysing the growth process. The starch in the endosperm thus serves as a food source for the new acrospire being formed in the germinated plant, hence the reduced carbohydrates in the malted FMSC (Table 2). The yield and functional properties of the FMSC differed depending on the pre-treatment used. Hydrothermal treatment caused a low yield of the SC due to the pearling of the SC. Due to the heat treatment during the kilning of malted and hydrothermally treated grains, the lightness of the FMSC was reduced due to the Maillard reaction [25]. Heating gelatinised the starch and thus reduced the cooked paste viscosity in the pre-treated SC matter compared to the native FMSC.

### 2.3. Nutritional Composition and Health Benefits of Finger Millet Seed Coat

The comparative nutritional profile of FM wholemeal flour, refined flour, native, hydrothermally treated, and malted seed coat of FM is presented in Table 2. The SC accounts for the amply higher nutrients in wholemeal flour. The protein of FM is said to be easily digestible, hence its common use as raw material for infant weaning foods [27]. The FMSC contains more proteins (12–13.6%) than wholemeal (8.7%) or refined flour (3.6%). This makes the SC a more nutritional food material. FM proteins are made up of albumins (8%), globulins (15%) and prolamins (35–50%) and contain the highest amount of methionine compared to other cereals [28]. With the high digestibility reported, the SC will find applications in various food products, from bakery to powdered products. The lipid content of FMSC is about 3 g/100 g (Table 2). The lipids in FM are mostly triglycerides, with oleic, linoleic, and palmitic acids as the main fatty acids. FM lipids have been reported to lower the risk of duodenal ulcers [6]. There is, however, a paucity of research on the fatty acid profile of FMSC lipids. With an increase in the lipid content of hydrothermally treated FMSC, there could be a change in the fatty acid profile. This warrants further studies. FMSC contains minerals such as calcium, iron, zinc, manganese, magnesium, and phosphorus (P). FM is especially rich in calcium and phosphorus, which are more concentrated in the seed coat fraction (711–830 mg/100 g). During germination, there is increased phytase activity leading to the phytate content due to the action of endogenous phytase in the grain, as well as exogenous phytase from yeast and sourdough fermentation, which can both release phytic acid complexed minerals. It is well known that phenolic acids function as antioxidants by giving electrons or hydrogen. Additionally, the oxidation of several food constituents, particularly fatty acids and oils, is inhibited by their stable radical precursors. The FMSC is quite low in carbohydrates (<20%) and high in dietary fibre and phenolic compounds. The hypoglycaemic, nephroprotective, hypocholesterolemic, and anti-cataractogenic properties of FMSC were proved in streptozotocin-induced diabetic rats on a 20% FMSC diet [29]. Conversely, Okoyomoh et al. [30] reported antidiabetic, antioxidant, hypoglycemic and nephroprotective properties of FMSC in streptozotocin-induced diabetic rats on a 40% FMSC diet. To achieve these protective effects, the authors noted that alkaline phosphatase, aspartate transaminase, and alanine transaminase serum levels were reduced.

Additionally, the content of thiobarbituric acid reactive compounds was dramatically reduced while the activities of catalase and superoxide dismutase were elevated [30]. Recently, the anti-ageing effect of FM has been documented [31]. Therefore, millet SC can be used as a functional component for the creation of functional foods for diabetics to gain positive benefits in preventing dyslipidaemia and the regulation of glucose homeostasis, thereby aiding in the management of diabetes and its co-morbidities [29]. A study on the hyperglycemia reduction in non-insulin-dependent adult diabetics revealed that pancreatic amylase and intestine α-glucosidase could effectively inhibit FMSC [32]. The reduced starch digestibility and subsequent postprandial glycemic response of FMSC can be attributed to its dietary phytates and polyphenols. The health-promoting qualities of FMSC can be ascribed to its phenolic compounds and dietary fibre.

#### 2.3.1. Dietary Fibre

The SC of FM contains more percentage of insoluble fibre (38.4–51%) than soluble fibre (1.2–6.4%). Dietary fibre is divided into several categories, including pectin, cellulose, lignin, and arabinoxylan. So far, the latter has gained the most research attention, as seen in the subsequent section.

##### Arabinoxylan

Arabinoxylan (AX) is a non-starch polysaccharide made of branched heteroglycans with a side chain composed of pentose sugars, arabinose, and xylose [28]. Recently, AX has gained popularity due to its health-promoting effects in treating diabetes and colon cancer. They are effective natural immunomodulators and prebiotics and are functional food ingredients [33]. The FM-AX has been successfully extracted using alkaline solutions and water. Water-extractable xylan extracted by Prashanth & Muralikrishna [28] revealed that the xylan was heat stable (up to 200 °C) and amorphous. Given that xylose was the predominant sugar found, the potential of the water-soluble xylan from FM seed coat for producing useful ingredients such as xylitol may be further explored [28]. On the other hand, insoluble AX (primarily made up of arabinose and xylose) was extracted from FMSC using saturated barium hydroxide and 1M potassium hydroxide. The AX had average molecular weights ranging from 40 to 1028 kDa [27,32]. According to recent research, a nutraceutical made from AX extracted from finger millet may alleviate obesity from a high-fat diet [34].

Arabinoxylans have also been reported to be a strong natural immunomodulator, specifically because of the hydroxycinnamic acids present in the bran [35,36]. This was demonstrated by extracting AX from FM-SMC using alkaline solutions: barium hydroxide Ba (OH)_2_ and potassium hydroxide (KOH). The AXs were subsequently purified and comparatively examined for their immune-stimulatory properties with the aid of murine lymphocytes and peritoneal exudate macrophages. The results indicate that significant macrophage activation, including phagocytosis and mitogenic activity, was shown by the arabinoxylans [37]. The AX extracted using Ba (OH)_2_ demonstrated above two-fold lymphocyte proliferation and macrophage phagocytosis than the AX extracted with KOH AX [35]. The ability of AX to induce macrophage phagocytosis is attributed to the ferulic acid content. Moreover, their findings demonstrated unequivocally that ferulic acid content directly correlates with the immunostimulatory action of AX [36].

As functional food ingredients, AXs influence weight regulation and reduction [13]. Sarma et al. [34] studied the effect of FM-AX on metabolic and gut bacterial abnormalities brought on by a high-fat diet (60% kcal from fat) in albino mice. Supplementing with FM-AX stopped weight gain from the HFD and reduced changes in hepatic inflammation, lipid accumulation, and glucose tolerance. This weight gain suppression is attributable to propionate—an end-product of FM-AX fermentation in the gut [38]. Specifically, propionate works by reducing cholesterol synthesis and suppressing the expression of HMG CoA reductase in the liver. Meanwhile, colonic administration of the drug has been found to increase GLP-1 and PYY in mouse jugular vein plasma [39]. Gene expression in the liver and white adipose tissue was improved. Additionally, AX supplementation enhanced the health of the ileum and colon and prevented metabolic endotoxemia. It also prevented metagenomic changes in the cecum.

##### Prebiotic Potential of Finger Millet Seed Coat Arabinoxylan

There is ongoing research into the bioconversion of agricultural by-products (such as AXs) into valuable functional macromolecules such as prebiotic xylooligosaccharides (XOS). Potential uses for the depolymerised xylan products, including xylose and XOS, include food, medicine, feed formulations, and agriculture. XOS, a non-digestible dietary component, is composed of xylobiose, xylotriose, and xylotetrose and has a lower degree of polymerisation (2 ≤ 10) than other sugars [40]. XOS has several health advantages, including decreasing the risk of colon cancer, lowering blood cholesterol levels, improving gut health and mineral absorption, encouraging probiotic growth, and helping those with Type 2 diabetes [41]. Palaniappan et al. [33] produced XOS from extracted water-soluble xylan of FMSC using enzymatic treatment, characterised it and tested the prebiotic efficacy of the derived XOS. There was a 72% yield of XOS from the enzymatic treatment of xylan, showing that this macromolecule can be produced on an industrial scale. Compared to commercial XOS and dextrose, the FMSC XOS was an efficient substrate for accelerating the growth rate and cell mass of L. plantarum [33]. Thereby validating the prebiotic potential of AX of FMSC.

#### 2.3.2. Polyphenols

Polyphenols are bioactive compounds found in plants with health-promoting properties when consumed [42]. Polyphenols impart anti-ageing, antioxidant and anti-hyperglycemic effects on humans when consumed [31]. Additionally, polyphenols stop the production of advanced glycation end products and restrict glucose absorption. The total polyphenol content of native FMSC ranged from 2356–11,200 mg/100 g [15,26], while the tannins and flavonoids were 2305 and 209 mg/100 g, respectively. The polyphenol content of native FMSC was higher than the wholemeal of FM (2300 mg/100 g) [33]. Conventionally, polyphenols are extracted from FM using the heat-reflux method—this entails refluxing the food material in acidified methanol. Balasubramaniam et al. [43] compared three polyphenol extraction methods: conventional heat-reflux, ultrasonication, and enzyme treatment (xylanase) plus ultrasonication (EU). They observed that enzymatic plus ultrasonic treatment enhanced phenolic yield by 2.3 times compared to the conventional method. However, yield with ultrasonication alone was on par with the conventional method. Total flavonoids increased by 1.4 times and 1.3 times, respectively, in ultrasonicated and EU extracts. Tannin concentrations rose dramatically (1.1-fold in ultrasonicated and 1.2-fold in EU extracts). The phenolic acid composition of FMSC, as identified in high-performance liquid chromatography and direct infusion electrospray ionisation mass spectrometry, includes proto-catechuic acid, naringenin, gallic acid, *p*-hydroxybenzoic acid, luteolin glycoside, ferulic acid, apigenin, syringic acid, *p*-coumaric acid, and *trans*-cinnamic acid, protocatechuic acid, gentistic acid, catechin gallates, epicatechin, caffeic acid, *trans*-cinnamic acid, kaempferol, and phloroglucinol [15,42,44].

### 2.4. Food Applications of Finger Millet Seed Coat

#### 2.4.1. Food Packaging

The environmental impact of plastic pollution is one of the problems that came with increased industrialisation. This has led to the demand for more sustainable and biodegradable packaging [45,46]. FMSC was recently used as a polysaccharide base for developing green food packaging material made from chitosan and squid gelatine [45]. The thickness of the film with 1% FMSC extracts increased by 65% compared with the film without bran extract. This increase is due to reduced space among the polymer chains in the film. Similarly, film opacity increased from 18% in the control sample to 68% in chitosan film with 1% FMSC. Thereby limiting the amount of light passing through the packaging. Meanwhile, a concomitant decrease in light transmittance (a result of increased thickness), water vapour permeability (62.5%) and absorption (due to polyphenol interaction in the film) was observed. The films with FMSC extract had higher antibacterial, antifungal and antioxidant activity than those without FMSC extract [45].

#### 2.4.2. Natural Antioxidants in Oils-Based Products

There is a rising demand for natural antioxidants from edible plant sources for food preservation and shelf-life extension. High-fat food items such as mayonnaise and salad dressings can be produced with natural antioxidants that are efficient, safe, and health-promoting to satisfy the rising demand for synthetic chemical-free products. The antioxidant capacity of FMSC polyphenols against lipid oxidation in mayonnaise was compared to a synthetic antioxidant (BHT). The study demonstrated that FMSC polyphenols (1.0 mg/g^1^) are more efficient than synthetic antioxidants (BHT) at preventing oxidative rancidity in a full-fat spread such as mayonnaise for seven (7) weeks at 4 °C [47]. In addition, the protein and total mineral content of the FMSC-enriched mayonnaise were markedly higher than its synthetic counterpart. Similarly, dose-dependent addition (200–1000 ppm) of FMSC polyphenols inhibited free-radical formation in peanut oil, thereby preventing oxidative rancidity and deterioration in oils during regular and accelerated storage of 7.8 weeks, respectively [43,48]. FMSC polyphenols were significantly effective in reducing primary and secondary oxidation products at 800 and 1000 ppm in oils stored for up to three months.

#### 2.4.3. Biscuits

Incorporating FM by-products in biscuits may provide natural health benefits, including calcium, iron, and zinc, to consumers. Krishnan et al. [12] studied the physicochemical and functional properties of native, malted and hydrothermally treated FMSC and then substituted them in wheat flour at 10 and 20% to make composite flour for biscuits (labelled as NFMSC, MFMSC and HTFMSC). The biscuits with significant nutrient variation from the control biscuit were 10% NFMSC, 20% MFMSC and 10% HT-FMSC biscuits. The protein content of MFMSC and HTFMSC increased by 8%, and carbohydrates decreased in the three samples by 6.8, 11.3 and 7.5%, respectively. There was a 3.5-, 7.6- and a 5-fold increase in the insoluble dietary fibre of the three biscuits. Dietary fibre, calcium, iron, and zinc of the biscuits made from composite flours increased. The 20% MFMSC biscuit had the highest increase in zinc and polyphenols at 26% and 70%, respectively [12]. The sensory scores showed that native and hydrothermally processed millet, the malted millet seed coat, was the best for making biscuits at 10% and 20%, respectively [12]. The instrumental colour profile of the biscuits was measured using the CIELab colour system, while the texture was assessed by testing the biscuits’ breaking strength using the three-point break technique. The addition of FMSC darkened and hardened the biscuits. Similarly, the average colour sensory score of the biscuits was 7, indicating an above-average likeness. Due to the roughness of the integrated fibre, the score for surface features was slightly lower, at 6.1. The biscuit had a texture score of 6.2, meaning it was less crisp but still maintained an acceptable rating of 7.6 for eating quality [7].

## 3. Possibilities and Viability of FMSC as a Source of Active Compounds

The nutrient profile of FMSC reported in this review, especially the bioactive compounds, albeit from a handful of studies, are comparable to those found in wheat bran—one of the most extensively-studied cereal brans in the literature. Moreover, the health-promoting properties of wheat bran bioactive compounds, including phenolic acids, arabinoxylans, alkylresorcinol and phytosterols, have been validated in animal and human studies [16]. The few in vivo studies, such as the strong inhibitory activity of FMSC towards α-glucosidase and pancreatic amylase [25], validate it as a functional food. The superior calcium content of FMSC, compared to other cereals, also makes it a unique functional ingredient, thereby justifying the need for its valorisation.

## 4. Conclusions

In contrast to its size, finger millet is a powerhouse of nutrients, especially the seed coat fraction. The process of FMSC extraction using the method of Shobana et al. [15] is simple and effective and requires less complicated technological operations than the method of Malleshi [14]. This does not mean the latter is less effective. It simply shows that during the production of decorticated FM grains, the by-product can be further utilised for food production, thereby reducing waste. The utilisation of FMSC is not limited to waste reduction because it accounts for the greater portion of nutrients in the cereal. FMSC is rich in calcium, phosphorus, polyphenols, tannins, and dietary fibre and less in fat. The nephroprotective, hypoglycaemic, anti-cataractogenic, and hypocholesterolemic effect of FMSC has been proven in animal studies and the anti-diabetic effect in animal and human studies. Regarding food use, there is a paucity of data—a clear indication of a huge research gap waiting to be tapped. Compared to major grains such as wheat, maize and rice, and other millets such as foxtail and pear millet, FM is still largely underutilised and cultivated at subsistence levels; it is no wonder the FMSC has lacked research attention, especially due to the literature data trend in the past decade. To contribute to the “zero hunger” goal of SDG by 2030, more research investments into developing FMSC-rich foods are required.

## Figures and Tables

**Figure 1 molecules-27-07837-f001:**
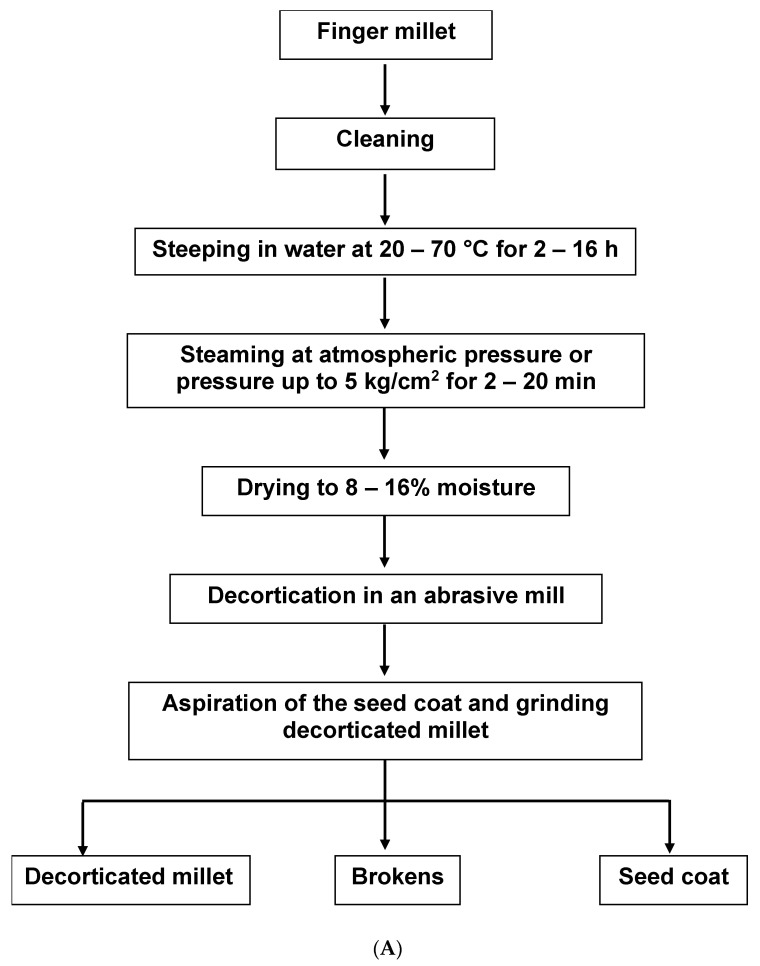
Extraction of finger millet seed coat. (**A**) Malleshi [14]; (**B**) Shobana et al. [15].

**Table 1 molecules-27-07837-t001:** Nutritional composition of finger millet and other cereal grains.

	Finger Millet	Pearl Millet	Foxtail Millet	Proso Millet	Wheat	Sorghum	Oat
Proximate (g/100 g)							
Moisture	13.1	12.4	11.2	11.9	12.8	12.4	8.2
Protein	7.7	11.6	12.3	12.5	11.8	10.6	16.9
Fat	1.5	5	4.3	1.1	1.5	3.5	6.9
Carbohydrate	79.7	67.5	60.9	70.4	71.2	72.1	66.3
Dietary fibre	19.6	11.3	2.4	9.0	12.5	6.7	10.6
Ash	2.7	2.3	3.3	1.9	1.5	1.6	3.2
Energy (kcal)	336	361	331	341	346	329	389
Minerals and trace elements (mg/100 g)							
Ca	350	42	31	14	30	13	54
Fe	3.9	8	2.8	0.8	3.5	3.36	4.7
Mg	137	137	81	153	138	165	177
P	283	296	290	206	298	222	523
Mn	5.94	1.15	0.6	0.6	2.29	0.78	45
Zn	2.3	3.1	2.4	1.4	2.7	1.7	3.97
Na	11	10.9	4.6	8.2	17.1	2	2
K	408	307	250	113	284	363	429
Vitamins (mg/100 g)							
Thiamin (mg)	0.42	0.33	0.59	0.2	0.45	0.33	0.76
Riboflavin (mg)	0.19	0.25	0.11	0.18	0.17	0.096	0.14
Niacin (mg)	1.1	2.3	3.2	2.3	5.5	3.7	0.96
Vit E	22	-	-	-	-	0.5	-
Folic acid (ug/100 g)	18.3	45.5	15	-	36.6	20	56

All values are expressed on a dry-matter basis. Sources: [3,4,5].

**Table 2 molecules-27-07837-t002:** Nutritional composition of wholemeal, refined flour and seed coat matter of finger millet.

Proximate Composition (g/100 g)	Native FMSC	Malted FMSC	Hydrothermally Treated FMSC	Wholemeal FM Flour	Refined FM Flour
Moisture	11.0–11.3	11	11	9.8	11
Fat	3.2–3.4	2.6	3.7	1.5	0.9
Protein	11.64–13.6	13.4	9.5	8.7	3.6
Carbohydrates	18.3–18.6	16.5	18.8	72	87
Ash	5.1–5.6	4.3	4.8	2.2	1.1
Dietary fibre					
Soluble	1.2–6.38	1.4	1.1	3.5	1.6
Insoluble	38.4–51.00	42.5	47.7	16.1	5.2
Total	39.6–57.38	43.9	48.8		
Minerals (mg/100 g)					
Calcium	711–830	707	864	321	163
Iron	5.9–6.5	5.5	7.5	3.3	0.33
Zinc	2.7–2.86	2.7	2.2	1.6	1.32
Phosphorus	369–526	253	344	201	106
Potassium	502	-	-	472	469
Phytochemicals (mg/100 g)					
Phytates	130–503	0.9	1.2	217	69
Polyphenols	2356–11,200	330	470	2300	1480
Tannins	2305			840	243
Flavonoids	209			136	106

Sources: [12,15,26], FMSC: finger millet seed coat.

## Data Availability

Not applicable.

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
