# Peer review of "Finger Millet Seed Coat—A Functional Nutrient-Rich Cereal By-Product"

_molecules, 2022, doi:10.3390/molecules27227837_

Round 1

Reviewer 1 Report

Today, the recovery of by-products generated in agri-food processes and the search for alternative raw materials as a source of active compounds, products to enrich the nutritional value of food or other applications is of great interest.

The article reviews the processes for obtaining finger millet seed coat (FMSC), its potential as a source of active compounds or its use in the development of foods or coatings. However, this review does not make a clear contribution to identify aspects that can promote further research.

Modifications and new contributions to the article are recommended:

1. In general, it is advisable to make a list of abbreviations or clearly define them before their use in the text. It is also necessary to unify the abbreviations and the concepts to which they refer. For example, on line 40 the abbreviation for FM seed coat is FMSC and on line 59 it is FM-SMC.

2. Restructuring of the text is recommended. After an introduction, in point 2.1., reference is made to the FMSC extraction processes. Point 3 refers to the pretreatment processes for this by-product. It is advisable to include this last point in the section corresponding to the processes to obtain the FMSC and indicate the advantages and disadvantages of the different processes and their effect on the composition of the nutrients and active compounds.

This may help to clarify the reason why in point 2.2. the comparison between native FMSC, malted FMSC and hydrothermally treated FMSC is included.

3. The article indicates that the FMSC are subjected to malting processes or hydrothermal treatment. According to the data indicated in table 2, these treatments affect the content of phytochemicals and other compounds of interest when compared to the native FMSC. Does it affect your profile, functionality and applications?

4. Table 1 does not indicate whether the contents of compounds are referred to 100 g of product, dry basis.... Clarify these data.

5. In line 59 it is advisable to include references that justify the effect of FM rich foods on diabetes.

6. In line 66 the initials SC appear without a previous indication of the concept to which they refer. In general, it is recommended to make a general revision of the text.

7. Line 89. Lipids present in FMSCs are not indicated. Are they the same as those identified in the FM?. Are they in the same proportion after the processes to which the FMSC is submitted?

8. Clarify the text of lines 138 to 141. At first they seem contradictory statements about the solubility of xylose.

9. Check writing of alkaline compounds in point 2.2.1.1.

10.   Clarify the text of lines 153 to 156.

11.   Line166. Unify criteria for references.

12.   Check the wording of lines 197 and 198.

13.   Line 218. Clarify what the acronym FMBE corresponds to.

14.   Lines 219 to 223. It is recommended to quantify the improvements obtained in the different properties evaluated in function of the quantity of FM bran extract added.

15.   Point 2.3.3. It is recommended to review the wording of lines 244 to 247. Clarify the type of FMSC that is used, percentage of substitution made and its correlation with the variation of nutrients in the biscuits

Indicate the scales and methods used to evaluate the colour, texture and other properties.

Finally, it is recommended to include a section about the possibilities and viability of FMSC as a source of active compounds compared to other sources of these functional compounds already on the market. Section that would allow to reinforce the conclusions and the justification for new research in the field of the valorisation of the FMSC.

Reviewer 2 Report

The manuscript entitled "Finger Millet Seed Coat – A Functional Nutrient-Rich Cereal By-Product" is well prepared and written in a compreensive laguage. It reviews the composition of Finger Millet Seed Coat and also the benefits for human health of the consum of this part of the cereal without putting apart  other possible industial uses of this material.

Despite the care of the authors in elaborating this manuscript there are some minor corrections to perform:

line 73: add "%" between" 5-10" and "w/v"

line 93: change "Finger millet" to FM; change "calcium and phosphorus and more concentrated in the" to "calcium and phosphorus and these minerals are more concentrated in the"

Figure 1: must be reformulated in order the two schemes have the same style, letter, size of the letter, boxes, etc.

Line 229: “demand for chemical-free products” change to “demand for synthetic chemical-free products”

In my opinion, after that minor corrections done the manscript is suitable for publication

Round 2

Reviewer 1 Report

The changes introduced have responded to the comments made in the first revision. These changes have notably improved the quality of the paper.